# Biowaste-Derived, Highly Efficient, Reusable Carbon Nanospheres for Speedy Removal of Organic Dyes from Aqueous Solutions

**DOI:** 10.3390/molecules27207017

**Published:** 2022-10-18

**Authors:** Bhavya Krishnappa, Vinay S. Bhat, Vimala Ancy, Jyotsna Clemi Joshi, Jyothi M. S, Maya Naik, Gurumurthy Hegde

**Affiliations:** 1Centre for Nano-Materials & Displays, B.M.S. College of Engineering, Bull Temple Road, Basavanagudi, Bengaluru 560019, Karnataka, India; 2Department of Civil Engineering, B.M.S. College of Engineering, Bull Temple Road, Basavanagudi, Bengaluru 560019, Karnataka, India; 3Department of Chemistry, AMC Engineering College, Bannerghatta Main Road, Bangalore 560083, Karnataka, India; 4Centre for Advanced Research and Development (CARD), CHRIST (Deemed to be University), Hosur Road, Bengaluru 560029, Karnataka, India; 5Department of Chemistry, CHRIST (Deemed to be University), Hosur Road, Bengaluru 560029, Karnataka, India

**Keywords:** kinetics, carbon nanospheres, oil palm leaves, adsorption

## Abstract

The current work explores the adsorptive efficiency of carbon nanospheres (CNSs) derived from oil palm leaves (OPL) that are a source of biowaste. CNSs were synthesized at 400, 600, 800 and 1000 °C, and those obtained at 1000 °C demonstrated maximum removal efficiency of ~91% for malachite green (MG). Physicochemical and microscopic characteristics were analysed by FESEM, TEM, FTIR, Raman, TGA and XPS studies. The presence of surface oxygen sites and the porosity of CNSs synergistically influenced the speed of removal of MG, brilliant green (BG) and Congo red (CR) dyes. With a minimal adsorbent dosage (1 mg) and minimum contact time (10 min), and under different pH conditions, adsorption was efficient and cost-effective (nearly 99, 91 and 88% for BG, MG and CR, respectively). The maximum adsorption capacities of OPL-based CNSs for BG were 500 and 104.16 mg/g for MG and 25.77 mg/g for CR. Adsorption isotherms (Freundlich, Langmuir and Temkin) and kinetics models (pseudo-first-order, pseudo-second-order and Elovich) for the adsorption processes of all three dyes on the CNSs were explored in detail. BG and CR adsorption the Freundlich isotherm best, while MG showed a best fit to the Temkin model. Adsorption kinetics of all three dyes followed a pseudo-second-order model. A reusability study was conducted to evaluate the effectiveness of CNSs in removing the MG dye and showed ~92% efficiency even after several cycles. Highly efficient CNSs with surface oxygen groups and speedy removal of organic dyes within 10 min by CNSs are highlighted in this paper.

## 1. Introduction

Dyes are non-biodegradable, lethal, possess dense aromatic structure and are long-lasting in the natural environment [1]. It is estimated that about 100,000 synthetic dyes are +available in the market and are used by textiles, paper, leather, cosmetics and many other industries. Globally 7 × 10^5^ to 1 × 10^6^ tons of dyes are produced every year, among which 10–15% of this reaches water bodies or soil as waste dyes [2]. Synthetic dyes are classified based on their properties and structure as anionic, cationic and non-ionic dyes. When compared to anionic dyes, cationic dyes are more toxic. Brilliant green (BG) and malachite green (MG) being cationic dyes, are used in a variety of applications including veterinary medicine, biological staining, paint and varnishes, tannery, parasite and fungus prevention in poultry feed and as dermatological agents [3,4,5]. MG dye is banned in many countries but is still used due to its cost effectiveness and the lack of proper alternatives [6,7]. An anionic azo dye, Congo red (CR) is used in industries such as cotton textiles, plastics, rubber, pulp and paper, rubber and cosmetics [8], despite its carcinogenic and mutagenic properties [9].

Dye-contaminated water reaching waterbodies has toxic compounds, intense color, and high bio-chemical and chemical oxygen demand [10]. In addition to the environment, the negative impacts of dyes on living creatures are significant and includes organ failure, vomiting, diarrhea, hypertension, respiratory ailments, and irritation to the skin and eyes in humans [11,12]. There is an immediate call for removal of dyes from water as it affects wellbeing by producing hazardous substances when they undergo reduction and oxidation reactions in water [13]. Physico-chemical technologies such as ion exchange [14], coagulation/flocculation [15], membrane filtration [16], oxidation [17], ozonation [18], the Fenton reagent technique [19], photocatalytic methods [20,21] and much more, are being used for dye removal from wastewater. Shortcomings of above-mentioned techniques include (i) enormous sludge production, (ii) being expensive, (iii) inefficiency in treating large volumes, (iv) high energy requirements, (v) toxic by-products, (vi) skilled supervision required, and (vii) complex operating practices [2,22,23,24]. Adsorption overcomes the disadvantages listed above and has emerged as a promising technology for removing dyes from wastewater because of its cost-effective, easy operational, and highly reliable characteristics, and is easily scaled up to industrial levels [25].

Much research has been done on different adsorbents available for dye removal, including activated carbon [26], zeolites [27], clay [28], agricultural waste [29], industrial byproducts [30], porous carbon nanospheres [31], metal organic framework [32,33,34] and many others. Nanomaterials have large surface area, pore volume and other enhanced surface properties which influence the rate and amount of adsorption [35]. On the other hand, agricultural biomass has high lignocellulosic content which provides a large number of active adsorbent sites helping in the adsorption of organic contaminants [36,37]. Some of the biowaste-based nanoparticles reported include titanium oxide nanoparticles used by loading them onto activated carbon prepared using watermelon rind (biowaste) in the removal of phenol red and Congo red dyes, and these showed high adsorption removal percentages of around 100% for both dyes [38]. Tripathi et.al, synthesized spherical-shaped carbon nanostructures using almond husk, a biowaste, in the removal of *p*-nitrophenol (dye pollutant) via adsorption, which had ~91% removal efficiency [39].

Kavci et al., used pinecone-based adsorbents treated with NaOH solution in the removal of malachite green dye, with around 80% efficiency [40]. Ahmad et al. used lime peel-based adsorbents treated with potassium hydroxide and removed ~94% of malachite green dye [40]. Arush Sharma s reported approximately 86 and 97% Congo red dye removal from *Cornulaca monacantha* stem (CS) and CS-based activated carbon prepared after treating with NaOH, respectively [41]. Muhammad Saif Ur Rehman et al., used biochar prepared from lignocellulosic bioethanol plant waste to remove brilliant green dye and reported a maximum of 77% efficiency [42]. To achieve maximum dye removal efficiency much research has used biowaste treated chemically (as mentioned above). Although removal efficiency is effective it requires skilled supervision to prepare adsorbents, which makes the process expensive. In addition, sludge disposal becomes an issue as this involves chemicals. To overcome these issues, in the present work oil palm leaves which are abundant, were used to synthesize carbon nanospheres that were further used as adsorbents in dye removal. A simple one-step pyrolysis process was adopted to synthesize adsorbents without involving any chemicals. The synthesized adsorbents were environmentally friendly and the treatment would be economical if used at a large scale at industrial levels. The adsorbents were highly competent in dye removal efficiency with a minimum adsorbent dose (1 mg) and short contact time (less than 10 min), as discussed in later sections of the article.

Waste biomass oil palm leaves (OPL) are abundant and an unexploited by-product of the palm oil industry of South-east Asian countries such as Malaysia, Indonesia, and Thailand, as well as in Africa and South America. OPL, a waste lignocellulose biomass encompasses lignin of 27.35%, α-cellulose of 44.53%, holocellulose of 47.7% and nearly 20.60% of extractives [43]. OPL-based nanoparticles have previously been used in a variety of applications, including cell imaging and guided drug delivery to cancer cells [44], supercapacitors [45], lipase immobilization [46] and many more. The lignocellulosic content of OPL makes it a rich source of carbon, which further enhances adsorptive properties of the material. 

We used OPL-derived carbon nanospheres (CNS) in the removal of model textile dyes from synthetic solutions. We attempted to synthesize CNSs from waste biowaste, OPL and use them as adsorbents to remove contaminants (BG, MG and CR) in an aqueous system. The rate of adsorption was optimized through key factors such as carbonization temperature of CNSs, amount of adsorbent, dye initial concentration, pH of the dye solution and contact period. Adsorption isotherms and kinetics models were derived using experimental data. A regeneration study was conducted to evaluate the efficacy of OPL-derived CNSs by rejuvenating them with the help of a washing solvent. Characterization of reused OPL-based CNSs obtained at the end of regeneration study was conducted. Despite the large amount of research on the adsorption of water contaminants, there is dispute over their use at an industrial scale and in the synthesis of adsorbent. Our work proposes an adsorbent derived from bio-waste that can be used for both cationic and anionic dye removal from aqueous systems.

## 2. Results

### 2.1. OPL 1000 Characterization

The morphology of the carbonized material seen using FESEM and TEM, showed distinctive features of CNSs prepared from pyrolysis. The particles appeared to be present as a conglomeration of spherical aggregates and not as discrete bodies (Figure 1a). The mushrooming of the spheres could be due to extended reaction times and cooling from the synthesis temperature to ambient temperature under controlled conditions. The UV-Vis spectra are mentioned in the Appendix A. The FESEM images of all the synthesized CNSs are shown in Appendix A of the Supplementary Section. No specific shape was formed at lower synthesis temperatures, while for 800 °C and 1000 °C we could see formation of spherical structures at the nano-scale. As the carbonization temperature was increased, we observed a decrease in particle size. The obtained CNSs showed a narrow size distribution for samples synthesized at 1000 °C (30–60 nm; Figure 1b). The presence of dangling bonds on the surface of CNSs was responsible for amalgamation of spheres, providing them with enhanced surface reactivity [37,41]. A distinctive solid spherical particle surrounded by a halo like structure can be seen in the TEM micrograph in Figure 1c, which supports the existence of such a feature. Energy dispersive X-ray spectroscopy (EDS) was used to identify the elements present in the carbon materials. The elemental composition of the precursor was compared with OPL 1000, which showed an increase of 20% carbon content (Table 1).

The thermal stability of CNSs was tested by TGA. The mass loss (%) as a function of temperature is depicted in Figure 1d. The TGA of CNSs were compared with that of precursor in three stages. Initial mass loss was due to removal of moisture and water content. Further mass loss from 250–450 °C was due to decomposition of cellulose and hemicellulose moieties in biomass. Further mass loss was gradual and due to decomposition of lignin. When this plot is compared with CNSs, the removal of moisture content occurred at ~100 °C, which accounts for the initial mass loss, while at higher temperatures, the mass loss was very insignificant as all lignocellulosic carbonization would have been completed during pyrolysis itself, except in case of the sample prepared at 400 °C, which contained undecomposed lignin and cellulosic moieties [42]. XRD patterns of all the synthesized CNSs with significant carbon content are shown in Figure 1e. A broadened hump at around ~23° followed by a sharp peak at 26° corresponds to the d_002_ plane in graphitic carbons. A broad peak at ~45° is ascribed to d_100_ plane found in turbostratic carbon [43]. Multiple minor peaks at 35.8, 39.2, 46.3 and 47.4 and 72° in all the samples of CNSs can be ascribed to the CaCO_3_ phase interconnected with the carbon matrix [44]. Regardless of the solid nature of the CNS, they exhibit poor graphitization as evaluated by Raman spectroscopy (Figure 1f). The Raman spectrum of the CNSs showed two major peaks at 1384 and 1595 cm^−1^, that are Raman shifts associated with the D-band and G-band respectively. The D-band is attributed to the presence of carbon structures with dangling bonds of the disordered graphite. This is indicative of a structural disorder in the carbon moiety. The G-band, on the other hand, corresponds to E_2g_ vibration of sp^2^ bonded carbons in a graphitic lattice. This is indicative of structural order [45,46]. The I_D_/I_G_ value, which reflects the relative intensity of D and G-bands, is used as an index to measure the magnitude of disorder or of graphitic nature. The I_D_/I_G_ value was found to be 0.52, 0.95, 1.07 and 1.13 for OPL 400, OPL 600, OPL 800 and OPL 1000 samples of CNSs, respectively, indicating increased dominance of defective carbon structures as the pyrolysis temperature was increased. A similar observation was observed in nanocarbons derived from waste onion peels [40]. The increasing disorder can be attributed to loosely formed particles which break into smaller crystallites when a large stress was induced in the form of heat. During carbonization, volatile gases escape from the precursor, resulting in the enrichment of a matrix rich in mainly carbon and oxygen. Hence, corresponding vibrations were identified in the FTIR spectrum (Figure 1g). The FTIR spectrum of the OPL precursor showed many functional groups, as expected. O-H stretching was identified by a peak at ~3413 cm^−1^. Stretching of -C-H at 2900 cm^−1^ in the precursor was absent in pyrolyzed CNSs. A peak at ~1591 cm^−1^ was formed due to -C=C- stretching, and a peak at ~1419 cm^−1^ corresponded to -C-H stretching in the carbon lattice, while a peak at ~1050 cm^−1^ was attributed to -C-O stretching.

The surface nature of the CNSs was also assessed from XPS. Two distinct peaks corresponding to carbon (C 1s) at ~285 eV and oxygen (O 1s) at ~532 eV were seen in the survey scan (Appendix A). Deconvolution of C 1s spectrum showed five peaks (Figure 1h). The more prominent peak at ~285 eV could be attributed to the sp^3^ carbon atoms (C-C) while the other peaks of relatively lesser intensities, centred around 284, 286, 286.8 and 287.8 eV, were due to the presence of C=C, -C-O, C-O-C and C=O groups, respectively [47]. These functional groups are thought be formed from groups of small molecules retained at the terminals, and to defects in the CNSs during carbonization [48]. Deconvolution of the O 1s spectrum yielded four subpeaks (Figure 1i). These peaks are generally attributed to double-bonded oxygen (531 eV), a C=O group (532.2 eV), an ether group (533.4 eV), single-bonded oxygen in carboxylic acid or ester groups (534.5 eV), or could also be due to chemisorbed water [49,50]. These surface functional groups in the form of dangling bonds play a pivotal role in efficient dye adsorption/removal. Overall, as the carbonization temperature was increased, the CNSs size decreased with increase in carbon content. Multiple peaks observed in OPL 400 disappeared at higher temperatures, while Raman spectra indicated the presence of defects in carbons as pyrolysis temperature was increased. FTIR spectra exhibited removal of many functional groups with carbonization, while the TGA plot indicated the stability of CNSs.

N_2_ adsorption-desorption isotherms were derived with the amount of gas adsorbed (cm^3^/g) plotted against equilibrium relative pressure (p/p_o_). Figure 2 depicts the isotherms of the OPL 1000 sample. The shapes of isotherms are similar to type IVa indicating a mesoporous nature. In the case of a Type IVa isotherm, capillary condensation is accompanied by hysteresis. A H4 type loop was observed, indicating presence of narrow mesopores. A Non-horizontal isotherm over the upper range of p/p_o_ indicates the presence of macropores (Figure 2a). The Brunauer–Emmett–Teller (BET) method was used to evaluate the surface area of porous carbons. OPL 1000 was found to have a BET-specific surface area of ~126 m^2^/g. The modified Kelvin equation proposed by Barrett, Joyner and Halenda (BJH) was used for pore size distribution and was found to be 2.3 nm (Figure 2b). A schematic illustration for our synthesis procedure is provided in Figure 2c.

### 2.2. Effect of Carbonization Temperature of CNSs

Carbonization temperature plays a vital role in the formation of CNSs. Surface characteristics, such as pore size and pore volume, are highly dependent on carbonization temperature because the carbohydrates and aromatic compounds must be pyrolyzed completely. This has a huge impact with respect to applications, for which optimization is essential. The effect of pyrolysis temperature of OPL on dye removal performance was assessed using MG dye, and the results are presented in Figure 3. With minimum contact time i.e., 5 min, OPL 400, OPL 600, OPL 800 and OPL 1000 showed a removal efficiency of around 65, 68, 74 and 81%, respectively (UV-Vis spectral graphs are shown in Appendix A of Supplementary Information (SI)). The trend was almost similar with higher contact times, and the obtained efficiency was in the range of 80 to 90%. As stated in introduction section, OPL has high lignocellulosic content, and its thermal behavior is distinct from that of lignin or cellulose due to its complex chemical constituents, requiring high temperature for its decomposition and complete carbonization. Hence, OPL carbonized at 1000 °C, possessed a better surface area and pore volume for greater removal efficiency of MG dye, with less work for future optimization factors.

### 2.3. Effect of Contact Time

After a feasible adsorbent was produced, the impact of contact period on removal of BG, MG and CR dyes was analyzed; the outcomes are presented in Figure 4. Results show that CNSs were efficient in adsorbing all three model dyes in less than 10 min; around 99% for BG, 90% for MG and 88% for CR. Respective spectral graphs are presented in Appendix A of SI. The removal of dyes from wastewater using CNSs requires numerous intricate interactions between the adsorbate and adsorbent (both chemisorption and physisorption). According to the most recent research, numerous mechanisms are at work during adsorption, some of which predominate depending on the system’s circumstances. Section 2.1 shows the surface functionalities present on CNSs via several characterization techniques. Oxygen-containing groups, such as C–O, C=O and –OH, on CNSs are the major functional groups responsible for adsorption of pollutants such as dyes [51]. The synthesized OPL was expected to have both acidic and basic surface groups containing oxygen, which could help in removal of both acidic and basic dyes within less time [52,53]. The pK_a_ values of BG, MG and CR are 5.36, 4.52 and 6.8, respectively, indicating the basic nature of BG and MG, and the acidic nature of CR. The acid groups on the CNS surface weakens the π-donor character of carbon atoms resulting in decreased CR adsorption. Nevertheless, there exist π- π interactions (face-to-face) between all the dyes and CNSs, which could synergistically lead to high adsorption within less time. The availability of extremely active adsorbent sites, and a high concentration gradient, contributes to the rapid removal of dyes in the early stages [11,54]. However, removal efficiency became constant after 40 min for all three dyes even when contact time studies were extended to 60 min. After the attainment of equilibrium time, the desorption of dyes was not significant, implying the stability of the adsorption process and showing the ability of OPL 1000 for use at large scale.

### 2.4. Effect of Adsorbent Dosage

The effect of the amounts of CNSs in the removal of BG, MG, and CR dyes was investigated by varying their dosage, while leaving all other components constant. Figure 5a–c shows the effect of CNSs content on performance, and Appendix A of SI show spectral graphs. It is clear from the graphs that dye removal efficiency of adsorbent increases with an increase in dosage for MG and CR. Removal efficiency of CNSs for the MG dye was 85, 90 and 95%, and for CR it was 83, 88 and 97% at 0.5, 1 and 1.5 mg, respectively. It must be noted that CNSs can remove almost 99% of BG dye irrespective of its dosage. Therefore, the dose of OPL-derived CNSs is directly proportional to the removal of MG and CR dyes. Taking account of the ease of measuring nanoparticles, 1 mg of CNSs was considered as optimum for the remaining BG adsorption studies. At higher dosages there are more empty adsorbent sites and a larger surface area, allowing dye molecules to penetrate the surface more easily [55]. However, as there was no big difference in percent removal of MG and CR dyes at 1 and 1.5 mg of adsorbent dose, to make the system more stable from an economic stance, 1 mg was fixed as the optimum dose for further studies.

### 2.5. Effect of Initial Dye Concentration 

The number of dye molecules and adsorbing sites available are vital factors in deciding the adsorptive property of any adsorbents. To investigate this, in our study the initial concentrations of BG, MG and CR dyes were 10, 15, 20, 25 and 30 µM. The results are shown in Figure 6a–e, respectively, as a function of time (respective spectral graphs are shown in Appendix A). The maximum removal efficiency of OPL 1000 at a 10 µM concentration was found to be about 99, 90 and 88%, whereas at a 30 µM concentration it was around 92, 76 and 64% for BG, MG and CR, respectively. At lower concentrations, the rate of adsorption was higher because the ratio of dye molecules to active adsorbent site was smaller, while at higher concentrations, the ratio was larger, resulting in pore saturation. At the saturation stage, dye molecules compete among themselves to bind to inner pores of adsorbent, resulting in reduced removal efficiency [56].

### 2.6. Effect of pH of the Dye Solution

As the surface charge of the adsorbent varies with pH, dye removal is also affected. pH studies were carried out at acidic, alkaline and native pH conditions. It can be observed from Figure 7 that OPL 1000 was successful in removing ~99% of BG dye in all pH conditions. It is evident from Figure 7 that percent removal of CR was high in acidic pH at around 99%, and for MG it was close to 87% at alkaline pH conditions. This may be due to the presence of a large number of H^+^ ions, which induce positive charges on the surface of the adsorbent creating difference in electric charge between the adsorbent surface and CR dye molecules, as well as developing an electrostatic force of attraction favoring the adsorption of negatively charged dye (CR) molecules [57]. On the other hand, at high basic pH conditions, a greater number of OH^−^ ions on the adsorbent surface makes it negatively charged and builds electrostatic attraction between its surface and positively charged MG dye molecules, resulting in high absorbance of the dye [12]. It must be noted that, though percent removal of CR dye at alkaline and at native pH conditions was less, the difference in removal efficiency with respect to the acidic condition was small, as evident in Figure 7. BG and MG, both are basic dyes that have aniline groups, but with a slight difference in alkyl groups attached to three benzyl groups. Due to this, the adsorption tendency was almost similar. Therefore, it can be concluded that OPL 1000 is efficient and effective in removing BG dye in all pH conditions, MG dye in alkaline pH conditions and CR dye in acidic pH conditions. Respective UV-Vis spectral graphs are presented in Appendix A of SI.

### 2.7. Adsorption Isotherms

Adsorption isotherm models provide information on mechanisms of adsorption, surface properties of the adsorbent, and the affinity and absorbability of adsorbent on to dye molecules [58]. To evaluate the adsorption capability of OPL 1000, experimental data were fitted to Freundlich, Langmuir and Temkin adsorption isotherm models. The Langmuir model assumes that the surface energy of the adsorbent is homogenous and has a finite number of active adsorbent sites leading to monolayer adsorption. Equations (1) and (2) represent the linear form of the Langmuir model and the dimensionless factor *R_L_*, respectively.
(1)$$\frac{C_{e}}{q_{e}}=\frac{1}{Q_{o}b}+\frac{C_{e}}{Q_{o}}$$
(2)$$R_{L}=\frac{1}{\left(1+bC_{0}\right)}$$

The amount of dye adsorbed per one gram of adsorbent at equilibrium (mg/g) is denoted by *q_e_*, *C_e_* is the adsorbate equilibrium concentration (mg/L), and *b* and *Q_o_* are Langmuir constants associated to rate of adsorption and capacity of maximum monolayer adsorption, respectively. *C_o_* indicates maximum initial solute concentration. The value obtained for *R_L_* shows adsorption is irreversible when *R_L_* = 0, linear when *R_L_* = 1, unfavourable when *R_L_* > 1 and favourable while 0 < *R_L_* < 1. The fitted line obtained for a plot of *C_e_*/*q_e_* versus *C_e_* is used to obtain the Langmuir constants *b* and *Q_o_*, which are the intercept and slope, respectively [58].

The Freundlich model describes the heterogeneity of the adsorbent surface and its allocation of active adsorbent high energy sites, which results in multilayer adsorption. The Freundlich model is expressed linearly in Equation (3)
(3)$$\log\;q_{e}=\log\;K_{F}+\frac{1}{n}\;\log\;C_{e}$$

Plotting log *q_e_* versus log *C_e_* yields values of *K_F_* and 1/*n*, where *K_F_* is the Freundlich constant linked to adsorption capacity in mg g^−1^, and 1/*n* is the adsorption strength in L mg^−1^ [59] The Temkin isotherm is used to determine the adsorbent-adsorbate relationship on the surface, and its linear form is given in Equation (4).
(4)$$q_{e}=B\;\ln K_{T}+B\;\ln C_{e}$$
where
$$B=\frac{RT}{b}$$
and *K_T_* represents the equilibrium binding constant in L g^−1^, *B* is adsorption heat in J mol^−1^ are Temkin constants are estimated by intercept and slope of the linear curve obtained from a plot of *q_e_* against ln *C_e_*. *T* denotes absolute temperature, which is 298 K, and *R* is the gas constant, equivalent to 8.314 J K^−1^ mol^−1^ [58]. The linear curves obtained when BG, MG, and CR dye adsorption results were fitted to Freundlich, Langmuir and Temkin adsorption isotherm models are shown in Figure 8. Table 2 presents the isotherm coefficients and correlation constants of all three dyes used in the study. The best fit obtained for the BG dye was the Freundlich isotherm model with an R^2^ value 0.9597. The best fit for MG was the Temkin adsorption model with an R^2^ value 0.9797, and the best fit for CR was the Freundlich adsorption isotherm model with an R^2^ value equal to 0.9613. The values of *Q_o_* describing the maximum adsorption capability of adsorbent for BG, MG and CR were 500.00, 104.16 and 25.77 mg/g, a trend well in agreement with the adsorption trend obtained in previous experiments, and substantiating the results with respect to involvement of surface oxygen groups and aromatic π-systems in the adsorption process. The obtained adsorption capacity of CNSs was compared with the available literature; the corresponding data is provided in Table 3. According to the comparison table, OPL-based CNSs were more effective than many nanomaterials, nanocomposites, and other biowastes and derivatives.

## 3. Discussion

Synthesis

Figure 2c depicts the formation of OPL-derived CNSs schematically. When the pyrolysis temperature is raised to 600 °C, the lignocellulosic breakdown of biomass produces volatile gases that, as they develop and escape through the biomass, form channels (B). Smaller pieces of nano carbons are generated as a result of pressure and temperature. Due to autogenerated pressure and temperature, the gases are driven into the previously constructed channels at 1000 °C, enlarging pore capacity. Furthermore, the biomass contains naturally embedded components, such as K, Na, and Ca, that augment ‘self-activation’. Initially, increasing the pyrolysis temperature provides more energy for gasification activities, and more carbon atoms are used to produce newer pores, increasing surface area and pore volume. As the temperature rises over the boiling point of potassium (762 °C), metallic potassium vapours develop, forming new porous structures (C). Although other gasification processes have been proposed for carbon consumption and pore creation, the following reaction may be more favourable since it is thermodynamically more spontaneous beyond 700 °C.
$$C\;\left(s\right)+{\;H}_{2}O\;\left(g\right)↔\mathrm{CO}\;\left(g\right)+H_{2}\left(g\right)$$

Kinetics of adsorption

Relocating of dye molecules from a concentrated solution onto the surface of adsorbent followed by adherence of dye molecules onto the pores of the adsorbent, as well as sorption itself, are the three major actions in the adsorption process [70]. The linear forms of pseudo-first-order, pseudo-second-order and elovich kinetic models were examined to analyse the adsorption mechanisms and rate limiting phases of BG, MG, and CR dye adsorption onto OPL 1000 CNSs, and are presented in Equation (5), Equation (6) and Equation (7), respectively
(5)$$\log\;(q_{e}-q_{t})=\log q_{e}-\frac{k_{1}}{2.303}t$$
(6)$$\frac{t}{q_{t}}=\frac{1}{k_{2}q_{e}^{2}}+\frac{t}{q_{e}}$$
(7)$$q_{t}=\frac{1}{\beta }\;\ln\left(\alpha \beta \right)+\left(\frac{1}{\beta }\right)\ln t$$
where *t* represents time in min, *q_e_* and *q_t_* are the adsorption capacities in mg/g at equilibrium and at time *t*, respectively, the pseudo-first-order rate constant is denoted by *k*_1_ (per min), the pseudo-second-order rate constant is symbolized by *k*_2_ (g mg^−1^ min^−1^), and relevant elovich constants *α* (mg g^−1^ min^−1^) and *β* (g mg^−1^) are calculated [70]. Graphs of kinetic models for BG, MG, and CR are shown in Figure 9. Table 4 summarises the rate constants, regression coefficients and *q_e_* values calculated so far. In all three dyes, the regression coefficient, R^2^, was strong for the pseudo-second-order kinetic model. Experimental *q_e_* values obtained were around 46.79, 41.09, 56.93 mg/g, and calculated *q_e_* values were 0.47, 90.90 and 61.72 mg/g for BG, MG and CR, respectively.

Although the difference between experimental and calculated *q_e_* values were quite large for BG and MG, their respective R^2^ value were 0.9998 and 0.9979 for the pseudo-second order-model. CR had a *q_e_* experimental value closer to the *q_e_* calculated value, and its R^2^ value was 0.9979 for pseudo-second-order kinetics. As a result, it can be inferred that adsorption for BG, MG, and CR dyes follow pseudo-second-order kinetics.

Reusability

Reusability of adsorbents is of great significance for a cost-effective system when millions of gallons of wastewater must be treated at the industrial level [71,72]. Removal efficiency of adsorbent for 10 µM MG dye was investigated for at least six cycles, and the results are given in Figure 10a,b. Nevertheless, before proceeding to a consecutive cycle, adsorbed dye was desorbed by washing with ethanol 2 to 3 times. It is important to note that removal efficiency of adsorbent remained at ~92%, even after six cycles. The stability or change in morphology of used/reused adsorbent is a key feature related to efficiency. After six cycles, reused adsorbent was characterized for its elemental composition and surface morphology by SEM and EDS; the results are shown in Figure 10c,d. EDS data revealed that ~85.77, ~10.51, ~3.27, and ~0.45% of carbon, oxygen, silica and calcium were present in reused CNSs, respectively. Freshly prepared OPL 1000 had a carbon content of ~86.92% and there was no significant change in carbon percentage even after six adsorption cycles. Therefore, OPL-derived CNSs are stable, effective and efficient in removing MG from industrial wastewater.

## 4. Materials and Methods

Materials & Characterization

Oil palm leaves were used as precursors for the preparation of CNSs. Finely powdered dried leaves were carbonized under inert condition at four different temperatures, i.e., 400, 600, 800 and 1000 °C for 1 h. The carbonized products were used for characterizations and further application studies. The detailed procedure for the synthesis of CNSs can be found in our earlier works [73,74,75,76]. The morphology of the prepared CNSs was confirmed from field emission scanning electron microscopy (FESEM; Apreo-FEI, Berkeley, CA, USA) and transmission electron microscopy (TEM; JEOL JEM-2010, Tokyo, Japan). Raman spectra of samples were measured using a Raman spectrometer (RAMAN force, Nanophoton, Japan). X-ray Photoelectron Spectroscopy (XPS) was used to investigate the surface chemistry of the CNSs (Thermo/ESCALAB 250XI, Berkeley, CA, USA). Infrared spectra were obtained by a Fourier Transform Infrared Spectroscopy (FTIR) technique for detection of the functional groups in CNSs (Perkin Elmer spectrum BX, Berkeley, CA, USA). CNSs were also subjected to heating from 26–1000 °C at 10.0 °C/min under a N_2_ atmosphere (thermogravimetric analysis) to assess their thermal stability (TGA7; Perkin Elmer, Berkeley, CA, USA).

Model dyes BG, MG oxalate and CR were supplied by Loba Chemie Pvt. Ltd. Mumbai, India. Dye stock and standard solutions were prepared using distilled water. Solutions of 1 M HCl and 1 M NaOH were used to alter pH. Ethanol was used as a washing solvent while conducting the regeneration study. The treated dye solution was segregated from dye adsorbed CNSs using a microprocessor-based centrifuge machine. pH variations were measured using a Hanna instruments pH meter (HI2210 pH meter). A UV-Vis Spectrophotometer (SHIMADZU UV-3100) was used to estimate the concentration of dyes in the treated dye solutions (absorbances at 625, 618 and 508 nm were measured for BG, MG and CR dyes, respectively). Scanning Electron Microscopy (Vega3 TESCAN) and Energy Dispersive X-ray Spectroscopy (TESCAN Vega III/Czech Republic) were used to assess the surface topography and elemental composition of reused CNSs, respectively.

Batch adsorption experiments

Synthesized CNSs were named after the pyrolysis temperature and referred to as OPL 400, OPL 600, OPL 800 and OPL 1000. Adsorption experiments were conducted for 60 min by adding 1 mg of CNSs synthesized at different temperatures into 10 mL of 10 µM MG dye solution (at native pH). The dye concentrations before and after adsorbent treatment were measured using a UV-vis spectrophotometer at a wavelength of 618 nm. Percent removal efficiency of CNSs was calculated using Equation (8)
(8)$$\mathrm{Removal efficiency}=\frac{\left(C_{o}-C_{e}\right)}{C_{o}}\times 100$$
where dye concentrations at initial and equilibrium conditions in mg/L are represented by *C_o_* and *C_e_*, respectively. Selected CNSs (OPL 1000) were used for optimization of contact time. BG, MG and CR dyes were used at native pH by adding 1 mg of adsorbent with 10 mL of 10 µM dye solution. Initial and final concentrations of dyes were analyzed at regular intervals and experiments were interrupted when saturation of removal efficiency was attained. To proceed with adsorbent dosage optimization, 0.5, 1.0 and 1.5 mg of adsorbents were used for 10 mL of 10 µM dye concentration at native pH. The studies were further extended to investigate the effect of pH of the target solution. The experiments were conducted as described above, but with 1.0 mg of OPL 1000.

## 5. Conclusions

OPL-derived CNSs were used as adsorbents in the removal of BG, MG and CR dyes from synthetic solutions. With just 1 mg of adsorbent and 10 min of contact time, OPL-derived CNSs showed excellent removal efficiency of ~99% for BG, ~90% for MG and ~88% for CR. CNSs were effective in removing BG dye in all pH conditions, whereas MG and CR dyes were removed more effectively in alkaline and acidic pH conditions, respectively. Adsorption data obtained are in good agreement with the Freundlich adsorption isotherm model for BG and CR dyes and the Temkin isotherm model for the MG dye. The adsorption capacity of OPL-based CNSs for BG was 500 mg/g, for MG was 104.16 mg/g and for CR was 25.77 mg/g. All three model dyes used followed pseudo-second-order kinetics reactions. OPL-based CNSs achieved approximately 92% of MG removal even at the end of several regeneration cycles. A synergy of pore size and pore volume, along with interactions between dangling oxygen functional groups of CNSs and dye molecules, enhanced removal efficiency. It can be concluded that abundant, readily available oil palm leaves used as precursors in the synthesis of CNSs through a one-step pyrolysis process without any chemical activation, makes them an excellent, eco-friendly, and cost-effective dye removal adsorbent.

## Figures and Tables

**Figure 1 molecules-27-07017-f001:**
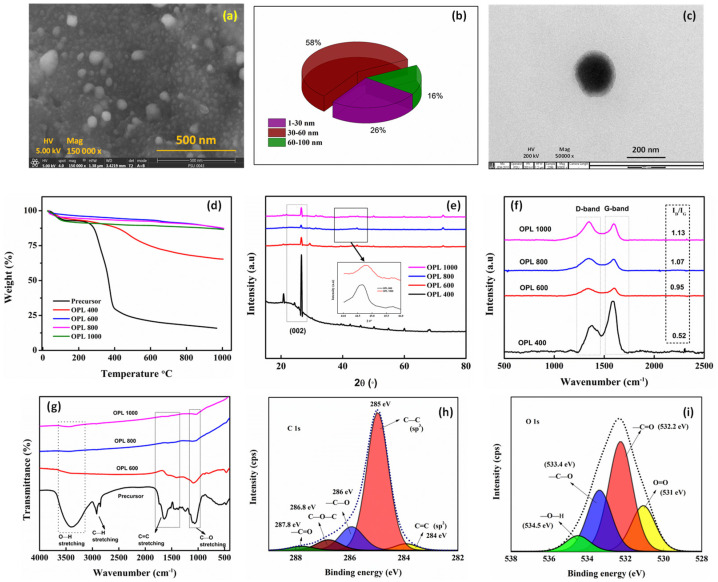
FESEM image of carbonized material synthesized at 1000 °C (**a**). CNSs size distribution obtained from FESEM image, depicted in a pie chart (**b**). TEM image of CNS (**c**). TGA of precursor and CNSs (**d**). XRD plot (**e**). Raman spectrum (**f**). FTIR spectrum of CNSs (**g**). Deconvoluted XPS spectra of C 1s (**h**). O 1s (**i**). The inset in 1e shows the magnified portion of OPL 800 and 1000.

**Figure 2 molecules-27-07017-f002:**
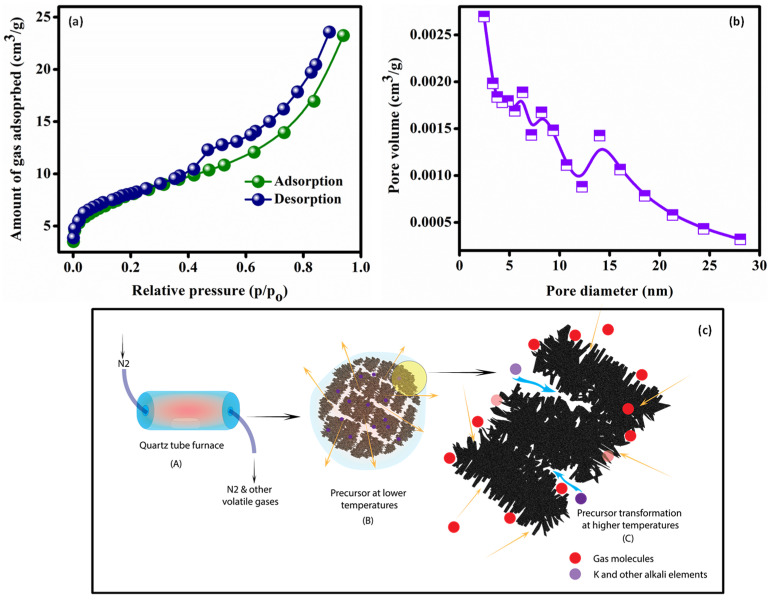
N_2_ adsorption desorption isotherms for the OPL 1000 sample (**a**). Pore size distribution for the OPL 1000 sample (**b**). Schematic illustration of the CNS synthesis process (**c**).

**Figure 3 molecules-27-07017-f003:**
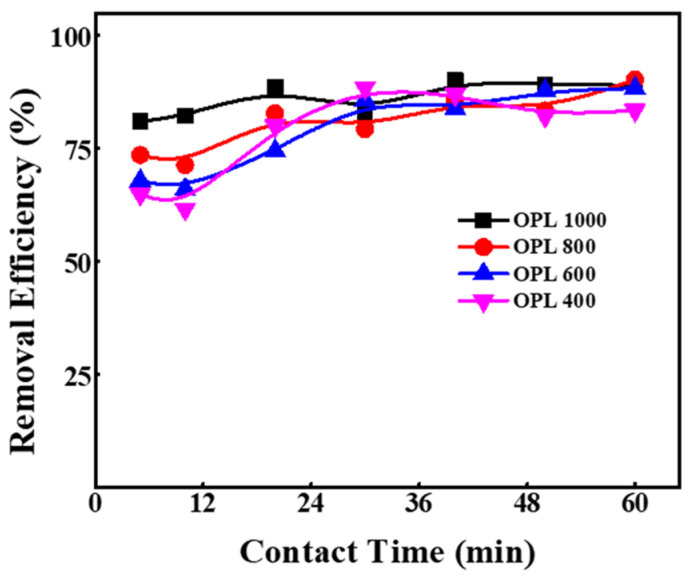
Effect of carbonization temperature of OPL-derived CNSs on removal of MG dye from an aqueous system.

**Figure 4 molecules-27-07017-f004:**
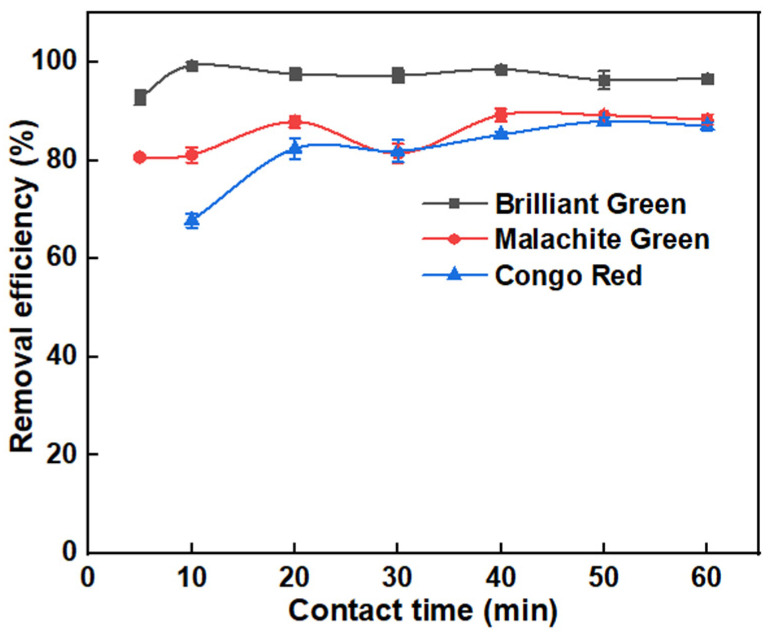
Effect of contact time on BG, MG and CR removal efficiency of OPL 1000 CNSs.

**Figure 5 molecules-27-07017-f005:**
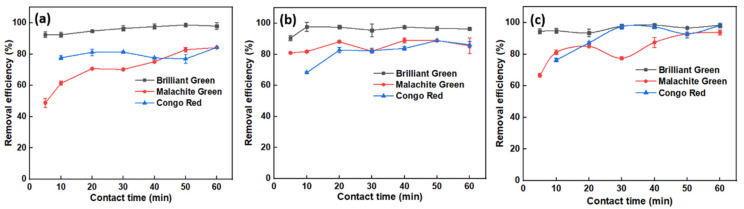
Impact of adsorbent dosage on the removal of BG, MG and CR dyes: (**a**) 0.5 mg, (**b**) 1 mg, (**c**) 1.5 mg.

**Figure 6 molecules-27-07017-f006:**
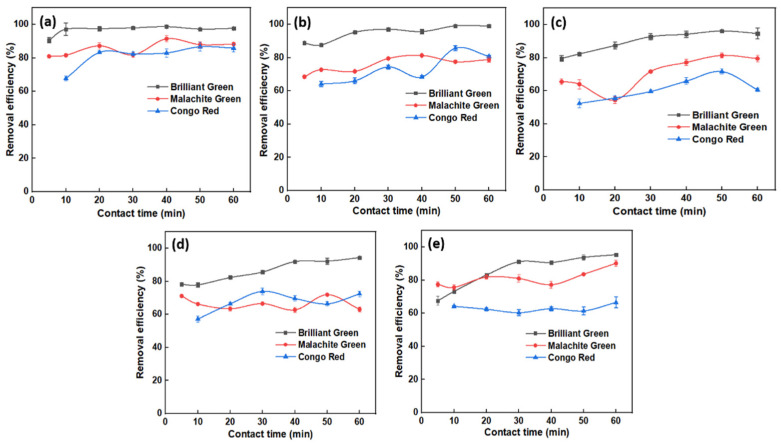
Effect on removal efficiency of OPL-derived CNSs when initial dye concentrations were (**a**) 10 µM, (**b**) 15 µM, (**c**) 20 µM, (**d**) 25 µM, and (**e**) 30 µM.

**Figure 7 molecules-27-07017-f007:**
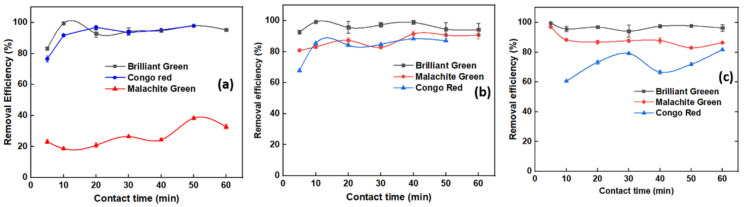
Effect of removal of BG, MG and CR dyes at different pH conditions. (**a**) Acidic, (**b**) native (**c**) basic.

**Figure 8 molecules-27-07017-f008:**
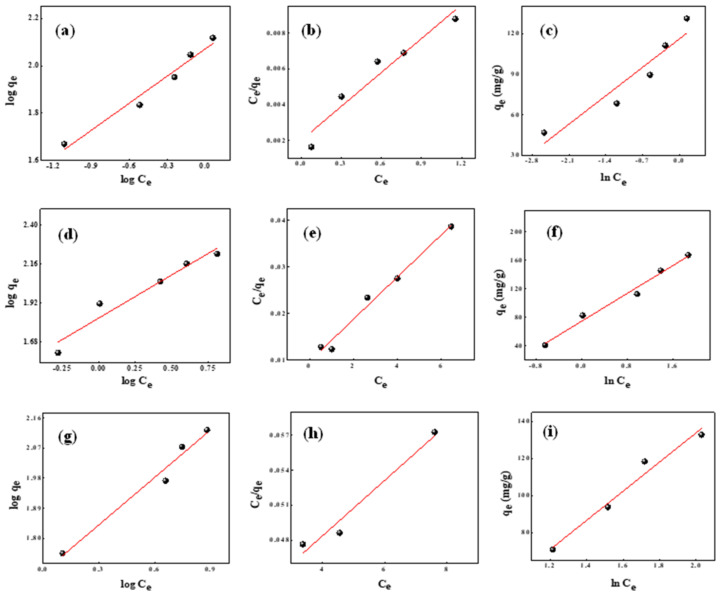
(**a**–**c**) Freundlich, Langmuir and Temkin isotherm plots for BG; (**d**–**f**) Freundlich, Langmuir and Temkin isotherm plots for MG; (**g**–**i**) Freundlich, Langmuir and Temkin isotherm plots for CR dyes.

**Figure 9 molecules-27-07017-f009:**
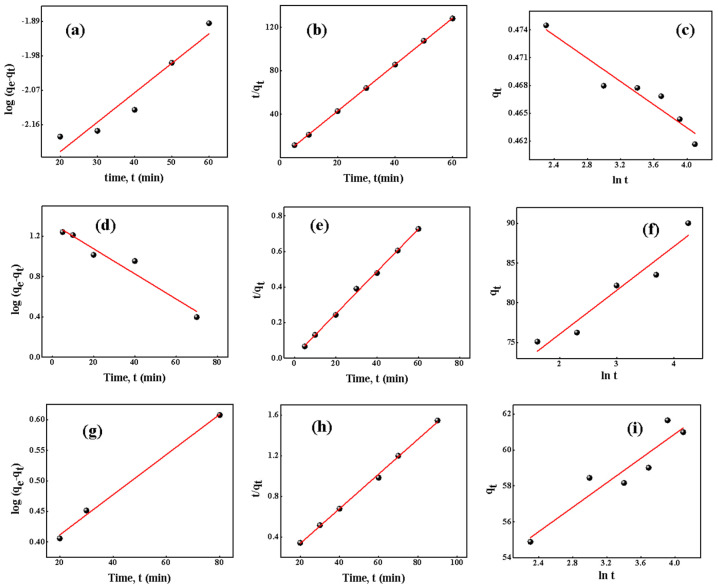
(**a**–**c**) Pseudo-first-order, pseudo-second-order and elovich models for BG. (**d**–**f**) Pseudo-first-order, pseudo-second-order and elovich models for MG. (**g**–**i**) Pseudo-first-order, pseudo-second-order and elovich models for CR.

**Figure 10 molecules-27-07017-f010:**
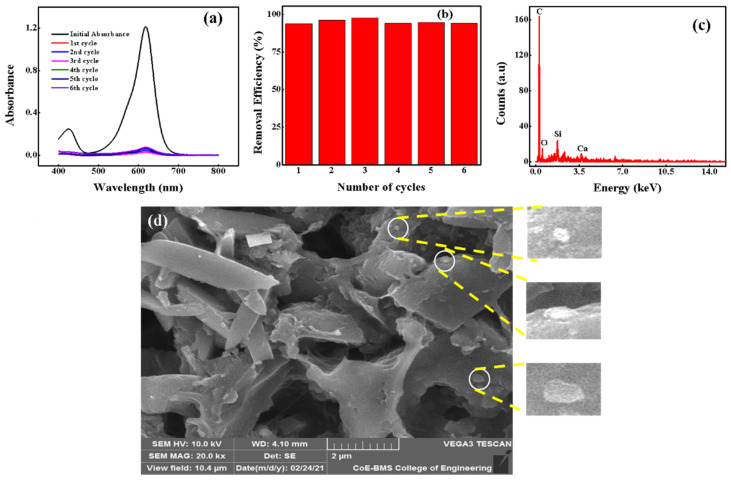
Reusability study of OPL-derived CNSs. (**a**) MG treated absorbance spectra in the reusability study. (**b**) OPL based CNSs showing number of adsorption cycles. (**c**) EDS plot of reused CNSs. (**d**) SEM image of reused CNSs after several cycles.

**Table 1 molecules-27-07017-t001:** Elemental composition of OPL precursor and OPL derived CNSs.

Material	Elemental Composition (Atomic Percentage) (%)			
	Carbon (C)	Oxygen (O)	Silicon (Si)	Calcium (Ca)
OPL precursor	67.2	32.34	0.34	0.12
OPL 800	84.63	13.88	0.69	0.62
OPL 1000	87.64	12.04	0.22	0.1

**Table 2 molecules-27-07017-t002:** Isotherm constants and correlation coefficients obtained for BG, MG and CR dyes.

Dyes	Langmuir Isotherm				Freundlich Isotherm			Temkin Isotherm		
	Q_0_	b	R_L_	R^2^	K_F_	1/n	R^2^	K_T_	B	R^2^
	(mg/g)				(mg/g)	(L/mg)		(L/g)	(J/mol)	
BG	500.00	0.555	6.55	0.9161	117.65	0.380	0.9597	47.643	30.05	0.8718
MG	104.16	2.133	0.05	0.9787	67.14	0.531	0.9263	4.6519	48.67	0.9797
CR	25.77	16.869	169.69	0.9344	50.00	0.472	0.9613	0.7388	78.91	0.9586

**Table 3 molecules-27-07017-t003:** Comparison of adsorption capacity of various adsorbents reported in the removal of dyes.

Sl. No.	Adsorbent	Contaminant	Adsorption Capacity, mg/g	Reference
1	Aniline propyl silica xerogel	CR	22.62	[59]
2	Banana peel	CR	1.727	[8]
3	Pineapple plant stem	CR	11.966	[60]
4	Rice husk ash	CR	7.047	[61]
5	Orange peel	CR	14	[62]
**6**	**OPL based CNSs**	**CR**	**25.77**	**Present work**
7	Graphene oxide cellulose bead composites	MG	30.09	[57]
8	Super-paramagnetic sodium	MG	47.84	[6]
9	activated sintering process red mud	MG	60.5	[63]
10	ZnO nanorod-loaded activated carbon	MG	59.17	[64]
11	Waste pea shells	MG	6.20	[65]
12	NaOH modified luffa aegyptica peel	MG	78.79	[66]
13	Cobalt ferrite silica magnetic nanocomposite	MG	75.5	[67]
**14**	**OPL based CNSs**	**MG**	**104.16**	**Present work**
15	Red clay	BG	125	[68]
16	Modified bambusa tulda	BG	41.67	[11]
17	Magnetite based nanocomposites	BG	252.17	[1]
18	Chemically modified areca nut husk	BG	18.21	[2]
19	Chemically treated Lawsonia inermis seeds powder	BG	34.96	[13]
20	Silver nanoparticles	BG	27.20	[25]
21	Modified pozzolan	BG	350.6	[69]
**22**	**OPL based CNSs**	**BG**	**500**	**Present study**

**Table 4 molecules-27-07017-t004:** Kinetic constants for all three models used in study.

Dyes	Pseudo First Order			Pseudo Second Order				Elovich Model		
	k_1_	q_e_	R^2^	k_2_	q_e_	q_e_^2^	R^2^	α	β	R^2^
	(min^−1^)	(mg/g)		(g/mg/min)	(mg/g)			(mg/g/min)	(g/mg)	
BG	1298.70	0.0927	0.8996	26.99	0.47	0.215	0.9998	9921.04	161.29	0.9025
MG	3333.3	3.7618	0.9315	001.10	90.90	8260	0.9979	7190.64	0.1811	0.9091
CR	6250	1.4129	0.9924	161252.42	61.72	3419.855	0.9979	3634.12	0.2935	0.8559

## Data Availability

Data are presented in the manuscript. Any other relevant specific data can be made available on request.

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
