# Peer review of "Biowaste-Derived, Highly Efficient, Reusable Carbon Nanospheres for Speedy Removal of Organic Dyes from Aqueous Solutions"

_molecules, 2022, doi:10.3390/molecules27207017_

Round 1

Reviewer 1 Report

It is an interesting work on the synthesis of biomass-based nanomaterials for environmental science application. The experiments are good-designed and the obtained results are convincible. Most of the conclusions are supported by the presented data. Still, there are some questions should be addressed carefully before the acceptance.

1.     It is necessary for the authors to provide a scheme on the synthesis process of CNSs through bio-waste.

2.     In the Introduction part, it is suggested for the authors to give more information on the novelty and significance of this work by comparing with other biomass-based dye adsorbents.

3.     In the SEM and TEM images of Figure 1, it is hard to see the text in the figure and the scale bars. In addition, in order to show the feasibility of the synthesis method on CNSs, it is suggested for the authors to provide a control experiment to adjust the size/morphology of CNSs.

4.     The inset histograms in Figure 10a is hard to see.

5.     The adsorption mechanism of CNSs towards organic MG dye should be discussed. It is better to provide a scheme to make it more clear.

6.     In Figure 10c, the SEM image indicates that after 6 cycles of adsorption of dyes, only very few CNSs can be found, and a lot of big sheets and aggregates were formed. What is the potential reason for that? More discussion is needed.

7.     English should be improved, and the reference format should be modified to fit the standards of this journal.

Reviewer 2 Report

The authors submitted a paper with the title “Bio-waste Derived, Highly Efficient, Reusable Carbon Nanospheres for a Speedy Removal of Organic dyes from Aqueous Solutions”. Here are my comments:

1.       There is much-activated carbon derived from biowaste and modified thermally and chemically to adsorb pollutants with much higher efficiency from aqueous media. The novelty and necessity of this work under question need to be clearly mentioned in the abstract.

2.       Authors should define the terms before directly using their abbreviations. “MG”, “BG”, and “CR” in the abstract.

3.       Authors should be consistent in using abbreviations. For example, the text used “Fig. X”, while in the captions they wrote “Figure X”.

4.       The Abstract is not well designed. Consider rewriting this part.

5.       In the introduction part there are references older than 15 years which are better to be replaced with recent relevant publications such as 10.1016/j.molliq.2018.07.108.

6.       Metal-organic frameworks as powerful, newly emerged adsorbents are missed from the listed adsorbents in line 68. Add recent papers from different research groups 10.1016/j.apsusc.2019.02.211 --- 10.1039/D0DT03824E---10.1016/j.jtice.2018.06.035

7.       The presented data should be repeated at least 3 times and error bars should be added.

8.       The adsorption mechanism (electrostatic/ π-π interaction / …) should be added to the graphical abstract and manuscript too.

The effect of ionic strength on the adsorption ability should be studied by the authors.

Reviewer 3 Report

This article is devoted to the study of the process of obtaining carbon nanospheres from biowaste for the rapid removal of organic dyes from aqueous solutions. The article is written in a clear and accessible language. Actuality and novelty is beyond doubt. I recommend improving the following points:

1. Abstract can be expanded.

2. In the introduction, you can quote: 10.3390/catal11080970.

3. Unification of figures throughout the article is required.

4. As a comparison, you can calculate and add the Elovich model for this system.

5. It is advisable to double-check the article for spelling and punctuation errors.

Round 2

Reviewer 1 Report

In this revised version, the authors made suitable modification on the manuscript and now the manuscript can be accepted for publication in current form.

Reviewer 2 Report

The manuscript is well-amended and is ready for publication.